# What factors contribute to the need for physical restraint in institutionalized residents in Taiwan?

Ching-Fang Chien[1,2,3‡], Ling-Chun Huang[1,2,3], Yang-Pei Chang[1,2,3], Chung-Fen Lin[4], Chih-Cheng Hsu[4,5,6,7]*, Yuan-Han Yang[1,2,3,8]*

1 Department of Neurology, Kaohsiung Medical University Hospital, Kaohsiung Medical University, Kaohsiung, Taiwan, 2 Neuroscience Research Center, Kaohsiung Medical University, Kaohsiung, Taiwan, 3 Department of Neurology, Kaohsiung Municipal Ta-Tung Hospital, Kaohsiung Medical University Hospital, Kaohsiung, Taiwan, 4 Institute of Population Health Sciences, National Health Research Institutes, Miaoli, Taiwan, 5 Department of Family Medicine, Min-Sheng General Hospital, Taoyuan, Taiwan, 6 Department of Health Services Administration, China Medical University, Taichung, Taiwan, 7 National Center for Geriatrics and Welfare Research, National Health Research Institutes, Taipei City, Taiwan, 8 Post-Baccalaureate Medicine, Kaohsiung Medical University, Kaohsiung, Taiwan

‡ This author worked as first author on this work.
* endlessyhy@gmail.com (YHY); cch@nhri.edu.tw (CCH)

**Data Availability Statement:** The data underlying the results presented in the study are available from National Health Research Institutes (NHRI),

## Abstract

### Background

In Taiwan, physical restraint is commonly used in institutions to protect residents from falling or injury. However, physical restraint should be used cautiously to avoid side effects, such as worse cognition, mobility, depression, and even death.

### Objectives

To identify the rate of physical restraint and the associated risk factors in institutionalized residents in Taiwan.

### Methods

A community-based epidemiological survey was conducted from July 2019 to February 2020 across 266 residential institutions. Among the estimated 6,549 residents being surveyed, a total of 5,752 finished the study. The questionnaires were completed by residents, his/her family or social workers. The cognition tests were conducted by specialists and a multilevel analysis approach was used to identify cognition/disability/medical history/special nursing care/BPSD risk factors for physical restraints.

### Results

Of the 5,752 included institutionalized residents, 30.2% (1,737) had been previously restrained. Older age, lower education level, lower cognitive function, higher dependence, residents with cerebrovascular disease, pulmonary disease, dementia, and intractable epilepsy, all contributed to a higher physical restraint rate, while orthopedic disease and spinal

Taiwan. The contact email is webmaster@nhri.edu.tw.

**Funding:** This study was financially supported by the Ministry of Health and Welfare, Taiwan (https://www.mohw.gov.tw); National Health Research Institutes (NHRI) (07D1-FRMOHW04), Taiwan (https://www.nhri.edu.tw/); the Neuroscience Research Center, Kaohsiung Medical University (KMU-TC110B03), Taiwan (https://www.kmu.edu.tw/). Data collection and data analysis were supported by NHRI. The funders did not influence the study design, data collection and analysis, decision to publish, or preparation of the manuscript.

**Competing interests:** The authors have declared that no competing interests exist.

cord injury were associated with a lower physical restraint rate. Furthermore, residents with special nursing care had a higher restraint rate. Residents with most of the behavior and psychological symptoms were also associated with an increased restraint rate.

## Conclusions

We studied the rate of physical restraint and associated risk factors in institutionalized residents in Taiwan. The benefits and risks of physical restraint should be evaluated before application, and adjusted according to different clinical situations.

## 1. Introduction

Applying physical restraint means restricting the patient's body or limbs with different means or devices, such as soft bands attached to the bed or chair [1]. The goal of physical restraint is to protect the patients from hurting themselves, falling from the bed or hurting other people. It is also used for patients who have essential tubes inserted into their body [2, 3], such as a nasopharyngeal tube, Foley tube or endotracheal tube, to prevent the patient from mistakenly removing these vital tubes. Patients with unclear consciousness, violent behavior or suicidal ideation may need physical restraint because of the above reasons. Patients with neuropsychiatric disease, such as dementia, who have behavior and psychological symptoms (BPSD), most commonly receive physical restraint, due to the symptoms of their disturbing behavior [3–6].

However, physical restraint can be harmful if patients are vigorous and struggling, increasing falls, fracture rate and the possibility of death [7]. A literature review from 1966–2006 reported that several cases of deaths occurred as a direct or indirect result of physical restraint, such as asphyxia [8]. Physical restraint can worsen a patient's incontinence due to immobility [9]. Decreased external stimulation and physical activity can also deteriorate the cognition and behavior of dementia patients [10, 11]. Emotional stress, including anger, anxiousness, regression and depression can occur in patients who are still conscious, which highlights the ethical problems of physical restraint. The most common reason for using physical restraint is to protect the resident or patient from falling. Between 1980 and 2000, studies showed that the method of physical restraint did not achieve its goal of protection as previously assumed, but instead increased poor outcomes in hospitals and nursing homes [12]. American government made the Nursing Home Reform Act to improve the problem, and due to its financial support discretion, the Act achieved the effect of reducing physical restraint rate [13]. The current consensus is to reduce the application of physical restraint to the lowest possible.

A study in Norway found that declined cognition, lower activity of daily living (ADL) performance, and the presence of aggressive behavior were major factors associated with the application of physical restraint in nursing home patients [14]. Hofmann et al. also highlighted similar factors, including lower ADL performance, severe cognition or mobility impairment, agitation, and previous fall and/or fracture. To the contrary, physical restraint could result in worse ADL, deteriorated cognition, and walking dependence in these patient groups [15].

A previous epidemiological study in Taiwan revealed that elderly patients with cardiovascular, neurological or skeletal disease are most likely to be admitted to an institution [16]. Over 80% of institutionalized residents had dementia [17]. These diseases have the potential to cause several of the reasons for physical restraint. Physical restraint affects institutionalized residents in both physical and psychological respects, but there have been only a few studies in Taiwan discussing the prevalence of physical restraint and the related risk factors in long-term care institutions. It is important to know whether the physical restraint applied was necessary

or could be deliberated in the case there was room for improvement? These issues motivated the present study.

Our study was set out to report the characteristics of institutionalized residents which are related to physical restraint in Taiwan, and to seek for restraint prevention and improve the quality of life in these groups of patients in the future.

## 2. Methods

### 2.1. Study subjects

From July 2019 to February 2020, a cross-sectional, community-based epidemiology study was conducted in 266 residential long-term care service institutions in Taiwan. All institutionalized residents were eligible, we excluded residents who cannot fulfill the survey. A total of 6,549 residents were surveyed and 5,752 residents finished the study. Patient demographic information, cognition and disability assessments, BPSD, medical history and special nursing care were recorded.

This study was approved by the institutional review board of the Research Ethics Committee of the National Health Research Institutes (EC1080502). All participants, or their legal representatives, provided written informed consent before entering this study.

### 2.2. Data collection

The survey was conducted by the National Health Research Institutes of Taiwan. The study included two stages. The first stage was conducted by well-trained interviewers from July 2019 to November 2019. Interviewers completed the survey to assess previous incidences of physical restraint. BPSD, ADL, instrumental activities of daily living (IADL), medical history and special nursing care before admission were also recorded during the case visit. A mini-mental state examination (MMSE) score of <26 was recognized as abnormal. If the MMSE was <26, a second stage evaluation was arranged for a further cognition survey.

The second stage was conducted by senior neurologists between December 2019 and February 2020. The neurologists evaluated the clinical dementia rating (CDR) [18] for institutional residents with an abnormal MMSE [19] or a medical history of dementia. All the interviewers or neurologists participated in a training course before the survey to ensure consistency and to reduce inter-individual bias in the evaluation.

### 2.3. Investigated outcome and variables

**2.3.1. Survey of physical restraint.** Physical restraint was defined in our study as a physical limitation of an individual's ability to freely move his or her limbs. A question was developed by the research team before the survey: Did you have physical restraint at the institution? Responses were 'yes' or 'no'. If the response was yes, the frequency of physical restraint was also recorded in the questionnaires.

**2.3.2. Evaluation of cognition and disability.** Psychometric tests were administered, such that cognitive function was evaluated by the MMSE [19], and global function was evaluated by the CDR [18]. We evaluated the disability level of institutional residents using ADL [20] and IADL [21]. The ADL and IADL were determined based on interviews, or by observation or testing by well-trained interviewers.

**2.3.3. Evaluation of dementia with behavior and psychological symptoms.** We adopted the Neuropsychiatric Inventory (NPI) [22] and symptoms of wandering, nighttime behavior, interference behavior, self-harm or suicide to evaluate BPSD for each recruited resident. At

least one caregiver living with the participant reported the patient's cognitive status and behaviors.

**2.3.4. Evaluation of medical history and special nursing care.** The medical history of the residents was recorded through a disease list, which was completed by the resident or his/her family. This list included hypertension, diabetes, orthopedic disease, eye disease, cerebrovascular disease, coronary artery disease, atrial fibrillation, cancers, pulmonary disease, digestive disease, genitourinary disease, dementia, psychiatric disorders, intellectual disability, cerebral palsy, Parkinson's disease, spinal cord injury, and intractable epilepsy. Responses were 'yes' or 'no'.

We evaluated special nursing care through a self-reported form, which was completed by residents or his/her family. The categories included nasogastric tube change, gastrostomy care, nasogastric tube feeding, tracheostomy care, ventilator use, sputum suction, oxygen therapy, saturation measurement, central venous catheter care, intravenous/intramuscular/subcutaneous drug administration, Foley tube change, change cystostomy, Foley care, intermittent catheterization, enema, digital disimpaction, jejunostomy care, drainage tube irrigation, wound change and dressing, bedsore wound care, pain management, passive range of motion, hemodialysis, and peritoneal dialysis. Responses were 'yes' or 'no'.

## 2.4. Statistical analysis

We performed statistical analyses using the Statistical Package for the Social Sciences (SPSS, version 18.0, IBM, Armonk, NY). We evaluated the demographic data of all residents and the presence of physical restraint. The participants were classified into two groups, namely with restraint and without restraint as categorical variables. We compared the differences in variables between these two groups. The chi-square and Student t-tests were used to determine the statistical differences among categorical variables and continuous variables, respectively. A $p$-value of $<0.05$ was considered to indicate a statistically significant difference.

## 3. Results

In total, we surveyed 266 care service institutions, including 143 welfare institutions for the elderly, 113 general nursing homes, and 10 veteran homes. The total completion rate at the institutions was 88.9%. Among the 6,549 residents being surveyed, a total of 5,752 finished the study. The total completion rate for the residents was 87.8%. The average patient age was 77.1 ±13.5 years and female gender accounted for 50.8% of the total population.

Among the 5,752 institutionalized residents, 1,737 (30.2%) presented as having previously been restrained. The differences in age, gender, education, MMSE, CDR, ADL, and IADL between the institutionalized residents who had been previously restrained and those who had not are shown in Table 1. Residents who were previously restrained were significantly older than those who were not restrained (mean age ± SD: 77.9 ± 13.1 vs 76.7 ± 13.6, $p = 0.002$). There was no significant difference in gender between the two groups. Lower education level was significantly associated with the probability of residents being restrained ($p<0.001$). Patients with lower MMSE scores had a significantly higher ratio of being restrained (mean ± SD: 13.0 ± 5.9 vs 17.5 ± 6.6, $p <0.001$). The percentage of residents with severe dementia (CDR 3) was higher in residents who had been restrained compared to the group not being restrained (79.3% vs 49.9%). Residents who had been restrained had significantly worse ADL and IADL compared to those who had not been restrained (for ADL, mean ± SD: 8.1± 17.4 vs 32.1± 33.8, $p < 0.001$; for IADL, 0.3± 0.7 vs 1.3± 1.8, $p < 0.001$).

The differences in medical history between institutionalized residents who had been previously restrained and those who had not been previously restrained are shown in Table 2. In a

**Table 1. Demographic characteristics of institutionalized residents with and without physical restraint.**

|  | Restrained | Without-restrained | p-value |
|---|---|---|---|
|  | N = 1737 (%) | N = 4015 (%) |  |
| Age (years) (mean ± SD) | 77.9 ± 13.0 | 76.7 ± 13.6 | 0.002*, |
| Age (n, %) |  |  | < 0.001* |
| ≤ 65 | 267 (15.4) | 791 (19.7) |  |
| > 65 | 1470 (84.6) | 3224 (80.3) |  |
| Gender (n, %) |  |  | 0.727 |
| Male | 861 (49.6) | 1970 (49.1) |  |
| Female | 876 (50.4) | 2045 (50.9) |  |
| Education level (n, %) | n = 1673 | n = 3919 | < 0.001* |
| Illiteracy | 721 (43.1) | 1397 (35.7) |  |
| Education 1–6 years | 574 (34.3) | 1402 (35.8) |  |
| Education ≥ 7 years | 378 (22.6) | 1120 (28.6) |  |
| MMSE (Mean ± SD) | 13.0 ± 5.9 | 17.5 ± 6.6 | < 0.001* |
| MMSE (n, %) | n = 257 | n = 1801 | < 0.001* |
| Normal (27–30) | 7 (2.7) | 200 (11.1) |  |
| Mild cognitive impairment (20–26) | 28 (10.9) | 485 (26.9) |  |
| Moderate cognitive impairment (10–19) | 152 (59.1) | 919 (51.0) |  |
| Severe cognitive impairment (0–9) | 70 (27.2) | 197 (10.9) |  |
| Dementia stage (n, %) | n = 1389 | n = 2633 | < 0.001* |
| Very early dementia (CDR 0.5) | 34 (2.5) | 349 (13.3) |  |
| Mild dementia (CDR 1) | 71 (5.1) | 481 (18.3) |  |
| Moderate dementia (CDR 2) | 183 (13.2) | 488 (18.5) |  |
| Severe dementia (CDR 3) | 1101 (79.3) | 1315 (49.9) |  |
| ADL (Mean ± SD) | 8.1 ± 17.4 | 32.1 ± 33.8 | < 0.001* |
| ADL (n, %) |  |  |  |
| Total independence (ADL 100) | 5 (0.3) | 186 (4.6) |  |
| Mild dependence (ADL 91–99) | 6 (0.4) | 100 (2.5) |  |
| Moderate dependence (ADL 61–90) | 39 (2.3) | 671 (16.7) |  |
| Severe dependence (ADL 21–60) | 166 (9.6) | 961 (23.9) |  |
| Total dependence (ADL 0–20) | 1521 (87.6) | 2097 (52.2) |  |
| IADL (Mean ± SD) | 0.3 ± 0.7 | 1.3 ± 1.8 | < 0.001* |
| IADL (n, %) |  |  |  |
| Normal (IADL 8) | 0 | 46 (1.2) |  |
| Mild dependence (IADL 6–7) | 4 (0.2) | 125 (3.1) |  |
| Moderate dependence (IADL 3–5) | 45 (2.6) | 624 (15.5) |  |
| Severe dependence (IADL 0–2) | 1688 (97.2) | 3220 (80.2) |  |

*$p < 0.05$; MMSE, mini-mental state examination; CDR, clinical dementia rating; ADL, activity of daily living; IADL, instrumental activities of daily living.

review of the medical history, residents who had been previously restrained had a higher percentage of cerebrovascular disease, pulmonary disease, dementia, and intractable epilepsy compared to those who had not been previously physically restrained. The percentage of patients with orthopedic disease and spinal cord injury was lower in the previously restrained group compared to the non-restrained group.

The differences in special nursing care in relation to restraint history in institutionalized residents are shown in Table 3. Residents with special nursing care had a significantly higher percentage of being previously restrained (71.9% vs 44.2%, $p <0.001$). Residents who had been

**Table 2. Medical history of institutionalized residents by physical restraint.**

| Medical history | Restrained | Without-restrained | p-value |
|---|---|---|---|
| | N = 1737 (%) | N = 4014 (%) | |
| Hypertension | 1023 (58.9) | 2391 (59.6) | 0.634 |
| Diabetes | 517 (29.8) | 1220 (30.4) | 0.633 |
| Orthopedic disease | 104 (6.0) | 328 (8.2) | 0.004* |
| Eye disease | 66 (3.8) | 150 (3.7) | 0.909 |
| Cerebrovascular disease | 596 (34.3) | 1213 (30.2) | 0.002* |
| Coronary artery disease | 249 (14.3) | 546 (13.6) | 0.460 |
| Atrial fibrillation | 37 (2.1) | 111 (2.8) | 0.163 |
| Cancers | 43 (2.5) | 120 (3.0) | 0.281 |
| Pulmonary disease | 282 (16.2) | 445 (11.1) | < 0.001* |
| Digestive disease | 239 (13.8) | 594 (14.8) | 0.304 |
| Genitourinary disease | 248 (14.3) | 565 (14.1) | 0.840 |
| Dementia | 540 (31.1) | 761 (19.0) | < 0.001* |
| Psychiatric disorder | 275 (15.8) | 564 (14.1) | 0.079 |
| Intellectual disability | 34 (2.0) | 70 (1.7) | 0.577 |
| Cerebral palsy | 11 (0.6) | 31 (0.8) | 0.570 |
| Parkinson's disease | 135 (7.8) | 258 (6.4) | 0.064 |
| Spinal cord injury | 8 (0.5) | 87 (2.2) | < 0.001* |
| Intractable epilepsy | 80 (4.6) | 117 (2.9) | 0.001* |

*$p < 0.05$; Values are presented as number (%).

restrained had a higher percentage of nasogastric tube change or gastrostomy care, nasogastric tube feeding, sputum suction, oxygen therapy, saturation measurement, Foley care, enema, drainage tube irrigation, wound change and dressing, bedsore wound care, and passive range of motion (ROM).

The association between having been previously restrained at an institutional residence and each BPSD type is shown in Table 4. Most BPSD types were significantly associated with being previously restrained (all $p < 0.05$), except for wandering ($p = 0.393$), language offensive behavior ($p = 0.323$), fear or anxiety ($p = 0.776$), and depression ($p = 0.468$).

## 4. Discussion

Our study concluded that 30.2% of 5,752 institutional residents in Taiwan had been previously restrained. Older age, lower education level, lower cognitive function (MMSE and CDR) and higher dependence (ADL and IADL performance) were associated with an increased physical restraint rate. We also found that cerebrovascular disease, pulmonary disease, dementia and intractable epilepsy contributed to a higher physical restraint rate, while orthopedic disease and spinal cord injury occurred less frequently in the previously restrained group compared to the non-restrained group. Furthermore, residents receiving special nursing care had a higher restraint rate and residents with most BPSD types were also associated with an increased rate of being restrained.

The findings regarding resident risk factors for the use of restraint in our study are consistent with previous studies. Hamers et al. [5] surveyed 265 nursing home residents in The Netherlands in 2004, and found that 49% were previously restrained. The average restraint duration was for about 3 months. The primary reason to apply physical restraint was to prevent falls and the resident risk factors were poor mobility, care dependency and risk of falling

**Table 3. Prevalence of special nursing care in relation to physical restraint in institutionalized residents.**

| Special nursing care | Having physical restraint | | p-value |
|---|---|---|---|
| | **Never** | **Yes** | |
| | **N = 4015 (%)** | **N = 1737 (%)** | |
| **Special nursing care (n, %)** | | | < 0.001* |
| No | 2239 (55.8) | 488 (28.1) | |
| Yes | 1776 (44.2) | 1249 (71.9) | |
| **Nasogastric tube change or gastrostomy care (n, %)** | 950 (23.7) | 896 (51.6) | < 0.001* |
| **Nasogastric tube feeding (n, %)** | 924 (23.0) | 869 (50.0) | < 0.001* |
| **Tracheostomy care (n, %)** | 120 (3.0) | 55 (3.2) | 0.719 |
| **Ventilator use (n, %)** | 26 (0.7) | 14 (0.8) | 0.507 |
| **Sputum suction (n, %)** | 428 (10.7) | 372 (21.4) | < 0.001* |
| **Oxygen therapy (n, %)** | 179 (4.5) | 102 (5.9) | 0.022* |
| **Saturation measurement (n, %)** | 223 (5.6) | 122 (7.0) | 0.031* |
| **CVC care (n, %)** | 0 | 1 (0.1) | 0.302 |
| **IV, IM, SC, IVD (n, %)** | 45 (1.1) | 24 (1.4) | 0.404 |
| **Foley tube change (n, %)** | 11 (0.3) | 1 (0.1) | 0.099 |
| **Change cystostomy (n, %)** | 40 (1.0) | 16 (0.9) | 0.790 |
| **Foley care (n, %)** | 479 (11.9) | 360 (20.7) | < 0.001* |
| **Intermittent catheterization (n, %)** | 63 (1.6) | 36 (2.1) | 0.178 |
| **Enema (n, %)** | 288 (7.2) | 191 (11.0) | < 0.001* |
| **Digital disimpaction (n, %)** | 250 (6.2) | 112 (6.5) | 0.751 |
| **Jejunostomy care (n, %)** | 23 (0.6) | 11 (0.6) | 0.784 |
| **Drainage tube irrigation (n, %)** | 2 (0.1) | 8 (0.5) | 0.002* |
| **Wound change and dressing (n, %)** | 200 (5.0) | 145 (8.4) | < 0.001* |
| **Bedsore wound care (n, %)** | 29 (0.7) | 36 (2.1) | < 0.001* |
| **Pain management (n, %)** | 52 (1.3) | 26 (1.5) | 0.544 |
| **Passive ROM (n, %)** | 357 (8.9) | 227 (13.1) | < 0.001* |
| **Hemodialysis (n, %)** | 78 (1.9) | 32 (1.8) | 0.798 |
| **Peritoneal dialysis (n, %)** | 1 (0.02) | 0 | 1.000 |

*$p < 0.05$; CVC, central venous catheter; IV, intravenous; IM, intramuscular; SC, subcutaneous; IVD, intravenous dripping; ROM, range of motion.

in the opinion of nursing staff. Kirkevold et al. [14] evaluated 1,926 patients in Norwegian nursing homes in 2005, and found that 45% of patients with dementia and 37% of those on regular wards had been previously restrained. Cognitive decline, poor ADL performance, and the presence of aggressive behavior were factors associated with the use of physical restraint. Huang et al. [2] evaluated 847 institutionalized residents in Taiwan in 2007, and found that 62% were previously restrained. They revealed that the reason for using physical restraint were primarily for preventing falls and tube removal. The resident risk factors were high dependence and the agreement of his/her family members or social workers. Younger primary caregivers contributed to facility-level risk factors for the use of physical restraint in this study. It is worth noting the overall lower physical restraint rate in our study compared to other studies. There are no special dementia care institutions in Taiwan. The decision for physical restraint is made by doctors and trained nurses in Taiwan. Permission in written form by residents or their family is needed after thorough discussion. Every use of physical restraint needs a cautious evaluation. Once applied, regular visiting, vital signs monitoring and reviewing the appropriate use after restraint application are regulated. Individual differences, such as genetic

Table 4. Having physical restraint of institutionalized residents by frequency of each BPSD type.

| BPSD type | Having physical restraint | | |
|---|---|---|---|
| | Never | Yes | p-value |
| | N = 4015 (%) | N = 1737 (%) | |
| No | 2621 (65.3) | 1052 (60.6) | < 0.001* |
| Yes | 1394 (34.7) | 685 (39.4) | |
| Wandering | 250 (6.2) | 98 (5.6) | 0.393 |
| Nighttime behavior | 558 (13.9) | 373 (21.5) | < 0.001* |
| Language offensive behavior | 467 (11.6) | 218 (12.6) | 0.323 |
| Physical aggressive behavior | 230 (5.7) | 138 (7.9) | 0.002* |
| Interference behavior | 296 (7.4) | 162 (9.3) | 0.012* |
| Resistance against care | 430 (10.7) | 269 (15.5) | < 0.001* |
| Delusion | 368 (9.2) | 202 (11.6) | 0.004* |
| Hallucination | 295 (7.4) | 176 (10.1) | < 0.001* |
| Fear or anxiety | 445 (11.1) | 197 (11.3) | 0.776 |
| Depression | 525 (13.1) | 215 (12.4) | 0.468 |
| Self-harm or suicide | 37 (0.9) | 38 (2.2) | < 0.001* |
| Repetitive behavior | 341 (8.5) | 214 (12.3) | < 0.001* |
| Attacks on items | 70 (1.7) | 46 (2.7) | 0.025* |
| Inappropriate or unclean behavior | 167 (4.2) | 118 (6.8) | < 0.001* |

*$p < 0.05$; Values are presented as number (%); BPSD, behavioral and psychological symptoms of dementia.

factors [23], resident disease severity and staff education may cause differences in the overall restraint rate. Furthermore, our study involved 266 different institutions, and the large number of surveyed samples could decrease the selection bias compared to studies focusing on fewer institutions.

Huang et al. [2] reported a higher physical restraint rate in patients with cerebrovascular disease and dementia. Given the common features of unsteadiness, easy falling and BPSD, dementia patients are at higher risk of being restrained [12]. Physical restraint should be eliminated in dementia patients where possible because they are vulnerable to undergoing functional decline, incontinence, emotional stress and developing pressure ulcers [15]. As average life expectancy is increasing, the prevalence of dementia also grows. Due to an increasing dependency ratio in Taiwan, many families may send demented or older patients to institutions. Family members might be reluctant to have their relatives been restrained. The patients certainly suffer. Physical restraint seemed not necessary in demented patients, nor the only solution for BPSD. Set a safer place, promote staff education, manage BPSD better and let patients feel comfortable are issues to avoid the vicious circle. Hardware facilities such as light, furniture; software such as nostalgia therapy, music therapy and religious healing were developed for above purposes.

With increased age, elderly people are prone to have cognition decline, dementia, multiple chronic disease, depression and delirium. Their resistance to external pressure are weaker than at younger age [24]. The use of physical restraint could be a stress, even results in fear and angryness in elderly people. The ego-integrity, good social support, family relationship, regular exercise habit and religion are ways to help people cope with stress and low mood. However, these helpful conditions are commonly lacking in institutionalized elderly people. A review in 2014 revealed that patients had a tendency to develop deteriorated ADL, cognitive impairment and more agitation after being restrained [15]. It was also found that untreated psychotrauma

in elderly could complicate with sleep disturbance, depression, suicide, and late-life delusional disorder [25].

Diverse manifestations of cerebrovascular diseases in institutionalized residents could lead to an increased rate of physical restraint, such as unsteadiness, consciousness disturbance [26], nasogastric tube insertion and Foley tube indwelling.

Our study further revealed that pulmonary disease and intractable epilepsy are positively associated with a higher physical restraint rate, while orthopedic disease and spinal cord injury were associated with a lower physical restraint rate. In residents with pulmonary disease or intractable epilepsy, special nursing care for sputum suction, oxygen therapy and saturation measurement are important. These are also factors which contributed to a higher physical restraint rate in our study, perhaps due to uncomfortable sensations which resulted in a lack of cooperation by the patient. A study involving intubated patients in 2010 [27], revealed that the absence of physical restraints were a major risk factor for unplanned extubation. The negative association between orthopedic disease and spinal cord injury on physical restraint is expected to be due to the patient's flaccid, muscle atrophy, immobility and bed-ridden status. Two previous studies showed less falling and a lower physical restraint rate in those who were totally bed-ridden compared to those residents who could still independently transfer themselves [28, 29]. We did not find a significant difference in the physical restraint rate in residents with psychiatric disease in our study.

Special nursing care for nasogastric tube change or gastrostomy care, Foley care, drainage tube irrigation, wound change and dressing, bedsore wound care, and passive ROM, were associated with a higher physical restraint rate in our study. This could perhaps be due to the need for complicated maneuvers, but this does not apply to all special nursing care. There was no significant difference in the physical restraint rate for residents with tracheostomy care, ventilator use and hemodialysis, which are all vital routes of insertion into the body. Severe dependence or clear residents who were fully cooperative is suspected to be the reason for this difference. Remove unnecessary tubes as possible could also decrease physical restraint applications.

Previous studies have highlighted that the decision of staff and the patients' family were partly responsible for using physical restraint [2, 5]. A cross-sectional study of 572 patients in South Africa in 2016 [30], revealed that 23% had been previously restrained. Fifty-nine doctors and 159 nurses were surveyed and <15% of the nurses reported that they have received training, while 36% of the doctors reported having received some guidance in the application of physical restraint. The process of prescription and indication for the use of physical restraint was generally poor. The manpower shortage could contribute to a higher possibility of physical restraint application [31]. Improving staff education and resources regarding the caring, legal and ethical aspects of physical restraint may be helpful.

A strength of the current study was that we included an updated and randomized sample as well as a large amount of institutions were included with comprehensive data on cognition, disability, medical history, and special nursing care. Two-step cognition tests were completed by neurologists and we recorded BPSD not only with NPI but also with broader evaluations, such as self-harm behavior. This study helped us to better understand the prevalence of physical restraint in institutions in Taiwan, and the main reasons for institutionalized residents being restrained.

There were some limitations to the present study. First, we did not include other forms of restraint than physical restraint of the limbs in this study. Second, this is a cross-sectional study, and we, therefore, cannot understand the impact of physical restraint on institutionalized residents. Third, 797 cases in our study failed to complete the survey; the retention rate was 87.8%. This might lead to bias in the estimation of the physical restraint rate and its causes,

however, it is still acceptable compared to other similar epidemiological studies [2, 5]. Additional studies focusing on the longitudinal follow-up of institutionalized residents who received physical restraint are recommended.

We analyzed data of physical restraint from institutionalized residents in Taiwan, and concluded that 30.2% of institutional residents in Taiwan had previously been restrained. Clinical characteristics including older age, lower education level, lower cognitive function, higher dependence, cerebrovascular disease, pulmonary disease, dementia and intractable epilepsy contributed to a higher physical restraint rate, while orthopedic disease and spinal cord injury were associated with a lower restraint rate. Furthermore, residents with special nursing care had a higher restraint rate, as did residents with most BPSD types. We also reviewed studies discussing physical restraint in recent years. After understanding these risk characteristics, we can embark on methods to decrease physical restraint use. The issue of physical restraint is important and warrants improvement. The results of this study do not stand for the opinions of the Ministry of Health and Welfare, Taiwan.

## Acknowledgments

We sincerely thank each of our partners for their great contributions, including the professional committee invited by the Ministry of Health and Welfare (MOHW), Taiwan and National Health Research Institutes, Taiwan, the trained raters, neurologists or geriatric psychiatrists, the members of Taiwan Dementia Society, institutional administrators and staff, local government for each site, and everyone contributing to the study. This article is the report of collective result from dementia research in Taiwan. The results of the manuscript do not stand for the opinions of MOHW, Taiwan.

## Author Contributions

**Conceptualization:** Chih-Cheng Hsu, Yuan-Han Yang.

**Data curation:** Ling-Chun Huang.

**Formal analysis:** Chung-Fen Lin.

**Project administration:** Yang-Pei Chang.

**Resources:** Chih-Cheng Hsu.

**Writing – original draft:** Ching-Fang Chien.

**Writing – review & editing:** Yuan-Han Yang.

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
