## [Decision Letter · Decision Letter 0]

16 Mar 2022

PONE-D-21-40533What factors contribute to the need for physical restraint in institutionalized residents in Taiwan?PLOS ONE

Dear Dr. Yang,

Thank you for submitting your manuscript to PLOS ONE. After careful consideration, we feel that it has merit but does not fully meet PLOS ONE’s publication criteria as it currently stands. Therefore, we invite you to submit a revised version of the manuscript that addresses the points raised during the review process.

We look forward to receiving your revised manuscript.

Kind regards,

Tai-Heng Chen, M.D.

Academic Editor

PLOS ONE

Journal Requirements:

Reviewers' comments:

Reviewer's Responses to Questions

**Comments to the Author**

1. Is the manuscript technically sound, and do the data support the conclusions?

Reviewer #1: Yes

Reviewer #2: Yes

2. Has the statistical analysis been performed appropriately and rigorously? 

Reviewer #1: No

Reviewer #2: I Don't Know

3. Have the authors made all data underlying the findings in their manuscript fully available?

Reviewer #1: Yes

Reviewer #2: Yes

4. Is the manuscript presented in an intelligible fashion and written in standard English?

Reviewer #1: No

Reviewer #2: Yes

5. Review Comments to the Author

Reviewer #1: The authors present an epidemiological study examining the factors that contribute to the need for physical restraint in institutionalized residents in Taiwan. The study is interesting but can be sharpened further. Please see my comments below:

1. A key concern is the reporting of the study. It will be helpful if the authors use a standardized reporting guideline to report the study. As it stands now, the study details are mixed. For instance, under study subjects, the authors report information regarding data collection. Please rectify these.

2. It remains unclear if the outcome measures used are standardized. If they are, it will be helpful to provide more details regarding the measures such as the Cronbach's alpha.

3. The exact statistical tests performed should be mentioned.

Reviewer #2: The reason to conduct the study is not strong enough. What the authors found in the study probably can be predicted and had been studied previously in other countries. The authors need to sharpen why they need to conduct the study, particularly in the context of improving the situation. physical restraint should be also viewed not only from physical or psyschological aspects, but also from socio-spiritual aspects. It needs to be discussed clearly in the background, results, and discussion.

6. PLOS authors have the option to publish the peer review history of their article (what does this mean?). If published, this will include your full peer review and any attached files.

Reviewer #1: No

Reviewer #2: **Yes: **Bayhakki

---

## [Author Response · Author response to Decision Letter 0]

21 Jul 2022

Reviewers' comments:

Reviewer #1: The authors present an epidemiological study examining the factors that contribute to the need for physical restraint in institutionalized residents in Taiwan. The study is interesting but can be sharpened further. Please see my comments below:

1. A key concern is the reporting of the study. It will be helpful if the authors use a standardized reporting guideline to report the study. As it stands now, the study details are mixed. For instance, under study subjects, the authors report information regarding data collection. Please rectify these.

Answer: 

The inappropriate content was removed. And the structure of paragraph was adjusted to meet the standardized reporting guideline. In addition, during the revision process, we found some minor data errors in the people number, so we also arranged correction. Sorry for the extra-burden on proofreading.

2. It remains unclear if the outcome measures used are standardized. If they are, it will be helpful to provide more details regarding the measures such as the Cronbach's alpha.

Answer: 

The outcome in this study (having physical restraint) was measured by a single question: were you having physical restraint at the institution; if the answer is yes, the frequency of physical restraint was also recorded (please see the manuscript line 131-135) (revise with tracked change: line 132-138). The content validity of this question and the entire questionnaire used in this study has been approved by an expertise committee before the survey. Because the investigated outcome (physical restraint) was measured by a single question, we cannot calculate related statistics such as Cronbach's alpha. 

3. The exact statistical tests performed should be mentioned.

Answer:

Please see the attached revised manuscript (line 168-176)(revise with tracked change: line 171-181) for the detailed description of statistical tests used in this study. 

Reviewer #2: The reason to conduct the study is not strong enough. What the authors found in the study probably can be predicted and had been studied previously in other countries. The authors need to sharpen why they need to conduct the study, particularly in the context of improving the situation. physical restraint should be also viewed not only from physical or psyschological aspects, but also from socio-spiritual aspects. It needs to be discussed clearly in the background, results, and discussion.

Answer: 

Thank you for the comment. We want to survey the physical restraint rate in institutions due to no common regulation in Taiwan. In the view of health care worker, we thought the rate may be higher than we think, and the result might help us to figure ways to improve the rate. We added socio-spiritual aspects on physical restraint in the text.

---

## [Decision Letter · Decision Letter 1]

18 Aug 2022

PONE-D-21-40533R1What factors contribute to the need for physical restraint in institutionalized residents in Taiwan?PLOS ONE

Dear Dr. Yang,

Thank you for submitting your manuscript to PLOS ONE. After careful consideration, we feel that it has merit but does not fully meet PLOS ONE’s publication criteria as it currently stands. Therefore, we invite you to submit a revised version of the manuscript that addresses the points raised during the review process.

We look forward to receiving your revised manuscript.

Kind regards,

Tai-Heng Chen, M.D.

Academic Editor

PLOS ONE

Journal Requirements:

Reviewers' comments:

Reviewer's Responses to Questions

**Comments to the Author**

1. If the authors have adequately addressed your comments raised in a previous round of review and you feel that this manuscript is now acceptable for publication, you may indicate that here to bypass the “Comments to the Author” section, enter your conflict of interest statement in the “Confidential to Editor” section, and submit your "Accept" recommendation.

Reviewer #1: (No Response)

Reviewer #2: All comments have been addressed

2. Is the manuscript technically sound, and do the data support the conclusions?

Reviewer #1: Yes

Reviewer #2: Yes

3. Has the statistical analysis been performed appropriately and rigorously? 

Reviewer #1: Yes

Reviewer #2: Yes

4. Have the authors made all data underlying the findings in their manuscript fully available?

Reviewer #1: Yes

Reviewer #2: Yes

5. Is the manuscript presented in an intelligible fashion and written in standard English?

Reviewer #1: Yes

Reviewer #2: Yes

6. Review Comments to the Author

Reviewer #1: Most of the comments previously raised have been addressed. As a minor comment, please report the study using an appropriate reporting guideline such as the STROBE.

Reviewer #2: Since elderly has many changes in the body, psycho-socio-spiritual aspects of elderly should be considered in studying treatment or intervention for them.

7. PLOS authors have the option to publish the peer review history of their article (what does this mean?). If published, this will include your full peer review and any attached files.

Reviewer #1: No

Reviewer #2: No

---

## [Author Response · Author response to Decision Letter 1]

20 Sep 2022

Reviewer #1: Most of the comments previously raised have been addressed. As a minor comment, please report the study using an appropriate reporting guideline such as the STROBE.

Answer: Thank you for your suggestion. We re-examined the content of this article according to the STROBE guidelines, the results are attached in word file. We also made several improvements of the content.

Reviewer #2: Since elderly has many changes in the body, psycho-socio-spiritual aspects of elderly should be considered in studying treatment or intervention for them.

Answer: Thank you for your suggestion. We have made relevant information searching, and added it to the manuscript. The contents are as below: 

With increased age, elderly people are prone to have a cognitive decline, dementia, multiple chronic disease, depression and delirium. Their resistance to external pressure are weaker than at younger age (Garrido P, De Blas M, Giné E, Santos Á, Mora F. Aging impairs the control of prefrontal cortex on the release of corticosterone in response to stress and on memory consolidation. Neurobiol Aging. 2012 Apr;33(4):827.e1-9. doi: 10.1016/j.neurobiolaging.2011.06.011. Epub 2011 Jul 27. PMID: 21794953) 

The use of physical restraints could be a stress, even resulting in fear and angryness of elderly people. The ego-integrity, good social support, family relationship, regular exercise habit and religions are ways to help people cope with stress and low mood. However, these helpful conditions are usually lacking in institutionalized elderly people. A review in 2014 revealed that patients had a tendency to develop deteriorated ADL, cognitive impairment and more agitation after being restrained (Hofmann H, Hahn S. Characteristics of nursing home residents and physical restraint: a systematic literature review. J Clin Nurs. 2014 Nov;23(21-22):3012-24. doi: 10.1111/jocn.12384. Epub 2013 Oct 11. PMID: 24125061) 

It is also found untreated psychotrauma in elderly could complicate with sleep disturbance, depression, suicide, and late-life delusional disorder (Flannery RB Jr. Restraint procedures and dementia sufferers with psychological trauma. Am J Alzheimers Dis Other Demen. 2003 Jul-Aug;18(4):227-30. doi: 10.1177/153331750301800408. PMID: 12955787)

---

## [Decision Letter · Decision Letter 2]

29 Sep 2022

What factors contribute to the need for physical restraint in institutionalized residents in Taiwan?

PONE-D-21-40533R2

Dear Dr. Yang,

We’re pleased to inform you that your manuscript has been judged scientifically suitable for publication and will be formally accepted for publication once it meets all outstanding technical requirements.

Kind regards,

Tai-Heng Chen, M.D.

Academic Editor

PLOS ONE

Reviewers' comments:

Reviewer's Responses to Questions

**Comments to the Author**

1. If the authors have adequately addressed your comments raised in a previous round of review and you feel that this manuscript is now acceptable for publication, you may indicate that here to bypass the “Comments to the Author” section, enter your conflict of interest statement in the “Confidential to Editor” section, and submit your "Accept" recommendation.

Reviewer #1: All comments have been addressed

Reviewer #2: All comments have been addressed

2. Is the manuscript technically sound, and do the data support the conclusions?

Reviewer #1: Yes

Reviewer #2: Yes

3. Has the statistical analysis been performed appropriately and rigorously? 

Reviewer #1: Yes

Reviewer #2: Yes

4. Have the authors made all data underlying the findings in their manuscript fully available?

Reviewer #1: Yes

Reviewer #2: Yes

5. Is the manuscript presented in an intelligible fashion and written in standard English?

Reviewer #1: Yes

Reviewer #2: Yes

6. Review Comments to the Author

Reviewer #1: Thanks to the authors for addressing the comments raised. No further comments from me. Looking forward to seeing your paper published. Thank you

Reviewer #2: (No Response)

7. PLOS authors have the option to publish the peer review history of their article (what does this mean?). If published, this will include your full peer review and any attached files.

Reviewer #1: No

Reviewer #2: No

---

## [Editor Report · Acceptance letter]

8 Nov 2022

PONE-D-21-40533R2 

What factors contribute to the need for physical restraint in institutionalized residents in Taiwan? 

Dear Dr. Yang:

I'm pleased to inform you that your manuscript has been deemed suitable for publication in PLOS ONE. Congratulations! Your manuscript is now with our production department. 

Kind regards, 

on behalf of

Dr. Tai-Heng Chen 

Academic Editor

PLOS ONE